# Testing the Accuracy of a Bedside Screening Tool Framework to Clinical Records for Identification of Patients at Risk of Malnutrition in a Rural Setting: An Exploratory Study

**DOI:** 10.3390/nu14010205

**Published:** 2022-01-02

**Authors:** Laura Alston, Megan Green, Melanie Nichols, Stephanie R. Partridge, Alison Buccheri, Kristy A. Bolton, Vincent L. Versace, Michael Field, Ambrose J. Launder, Amy Lily, Steven Allender, Liliana Orellana

**Affiliations:** 1The Global Obesity Centre, Institute for Health Transformation, Deakin University, Geelong, VIC 3220, Australia; Melanie.nichols@deakin.edu.au (M.N.); steven.allender@deakin.edu.au (S.A.); 2Research Unit, Colac Area Health, Colac, VIC 3250, Australia; mgreen@cah.vic.gov.au (M.G.); ABuccheri@cah.vic.gov.au (A.B.); MField@cah.vic.gov.au (M.F.); ajlaunder@deakin.edu.au (A.J.L.); amy.lilly@outlook.com.au (A.L.); 3Deakin Rural Health, School of Medicine Faculty of Health, Deakin University, Geelong, VIC 3220, Australia; vincent.versace@deakin.edu.au; 4School of Health Sciences, Faculty of Medicine and Health, The University of Sydney, Camperdown, NSW 2006, Australia; stephanie.partridge@sydney.edu.au; 5Engagement and Co-Design Hub, The University of Sydney, Camperdown, NSW 2006, Australia; 6Prevention Research Collaboration, Charles Perkins Centre, Sydney School Public Health, The University of Sydney, Sydney, NSW 2006, Australia; 7Institute for Physical Activity and Nutrition, Deakin University, Geelong, VIC 3220, Australia; kristy.bolton@deakin.edu.au; 8Biostatistics Unit, Faculty of Health, Deakin University, Geelong, VIC 3220, Australia; l.orellana@deakin.edu.au

**Keywords:** rural, malnutrition risk, malnutrition screening, rural health services

## Abstract

This study aimed to explore the diagnostic accuracy of the Patient-Generated Subjective Global Assessment (PG-SGA) malnutrition risk screening tool when used to score patients based on their electronic medical records (EMR), compared to bedside screening interviews. In-patients at a rural health service were screened at the bedside (*n* = 50) using the PG-SGA, generating a bedside score. Clinical notes within EMRs were then independently screened by blinded researchers. The accuracy of the EMR score was assessed against the bedside score using area under the receiver operating curve (AUC), sensitivity, and specificity. Participants were 62% female and 32% had conditions associated with malnutrition, with a mean age of 70.6 years (SD 14.9). The EMR score had moderate diagnostic accuracy relative to PG-SGA bedside screen, AUC 0.74 (95% CI: 0.59–0.89). The accuracy, specificity and sensitivity of the EMR score was highest for patients with a score of 7, indicating EMR screen is more likely to detect patients at risk of malnutrition. This exploratory study showed that applying the PG-SGA screening tool to EMRs had enough sensitivity and specificity for identifying patients at risk of malnutrition to warrant further exploration in low-resource settings.

## 1. Introduction

Malnutrition has been estimated to affect 20–60% of in-patients on acute hospital wards globally [1,2,3,4,5,6,7,8]. Screening for malnutrition is the first and critical step in identifying and treating patients who are at risk, as it often presents as an ‘invisible’ condition, affecting patients across the spectrum of weight status [9].

There are multiple validated tools to identify malnutrition risk at the patient’s bedside [5,9,10,11]. Existing tools are designed to be used and completed by healthcare staff who do not have nutrition-specific training to identify patients who are at risk and require further dietetic assessment, but uptake is not always optimal [3,9]. Low uptake of malnutrition screening has been hypothesized to be due to the already large administrative burden on nursing or other staff who are expected to undertake malnutrition screening on admission [2,3,12].

Electronic medical records (EMRs) are steadily replacing paper-based medical records in developed countries [13]. Since the 1990s, technology has rapidly advanced and become cheaper and therefore EMRs have been more widely integrated into hospitals [13]. Algorithms within EMRs have been used to flag issues to clinical staff (such as patient drug allergies), reducing errors and enhancing physician compliance with allergy prevention protocols [14]. Despite the potential for EMRs to improve care, there has been minimal research on how EMRs can be used to improve care in other fields, such as allied health [14], particularly for malnutrition.

Previous malnutrition risk screening research has utilised clinical dietitians to undertake paper-based record audits to extract data and make clinical judgements on the level of nutrition risk within an in-patient population [3,12,15,16]. Alston et al. in 2020, published the first study applying the Patient-Generated Subjective Global Assessment (PG-SGA) to EMRs as a screening framework to estimate the proportion of in-patients at risk of malnutrition in a rural setting [17]. EMR data is collected by a range of health professionals (including dietitians) during the hospital patient’s admission. These records were retrospectively screened to assign a malnutrition risk score, guided by the PG-SGA. The study found that most in-patients (77%) had indicators of malnutrition risk in their EMRs, a prevalence that could not be adequately addressed with the current level of dietetics resourcing available in rural settings [17].

In Australia, rural populations experience poorer health relative to metropolitan counterparts [18,19,20] with a lack of dietary interventions documented in these rural areas [21]. Malnutrition has been shown to follow a similar pattern to other health inequities, with a substantially higher risk documented in rural settings [17]. This is of particular concern due to the long-term healthcare workforce deficiencies in rural areas, even in developed countries such as Australia and Canada [16,17,22]. Given the higher risk in these rural areas, and poor uptake of malnutrition screening globally; more effective, less onerous, and potentially automated ways of screening patients for malnutrition need to be investigated.

The PG-SGA, a validated malnutrition screening tool which guides the triage of nutrition intervention, when used as a bed-side screening tool, it has been validated and shown to have high reliability [23]. It can also be used by health professionals other than dietitians to prioritise referrals for specialised review [23]. However, the accuracy of traditional bed-side tools, such as the PG-SGA, when used to guide review of the text in EMRs has not been investigated.

The aim of this exploratory study is to investigate the accuracy of a malnutrition risk score obtained by applying the PG-SGA screening instrument to the clinical text in EMRs, relative to the PG-SGA instrument applied at the bedside, in identifying rural in-patients at an increased risk of malnutrition.

## 2. Materials and Methods

### 2.1. Study Design

Ethics approval was granted by the Deakin University ethics committee and a letter of confirmation was received from the Barwon Health ethics committee. This pilot exploratory study was undertaken among in-patients on the acute admissions ward, at a rural hospital in Victoria, Australia. Over a 12-week period, a sample of eligible patients were screened for malnutrition risk at the bedside by a member of the research team, following standard protocols for the PG-SGA [23]. Eligible patients were identified from in-patient admissions lists. Patients were eligible if they were on day 2 or more of admission (to enable adequate detail in medical notes), over the age of 18 years, able to give informed consent and not pregnant or breastfeeding. Exclusion criteria included those who were pregnant/breastfeeding, paediatric patients (<18 years) or those who were deemed ineligible to provide consent by the patients’ medical team. The same patients’ EMRs were then independently screened by other members of the research team (who were blinded to the bed-side screen score), retrospectively, using the PG-SGA.

The PG-SGA malnutrition screening tool used in this study [24], is typically used at the bedside to identify whether or not patients are at risk of malnutrition and in need of assessment by a clinical dietitian [23]. The tool includes items regarding level of oral intake, recent weight loss, age, conditions, fever, steroid medications, nutrition impact symptoms (poor appetite, nausea, constipation, mouth sores, no taste/poor taste, swallowing difficulties, pain, vomiting, diarrhea, dry mouth, sensitivity to smells and early satiety), and level of activities and function [23]. The information collected is summarised in a score with range 0–36; and values >4 are considered ‘at risk’ of malnutrition indicating a need or high need for dietetic assessment to address malnutrition (>9 indicates a ‘critical need’) [24]. The PG-SGA was used to interview patients at the bedside following standardised protocols [23]. The researchers selected the PG-SGA tool to explore in this study due to the large amount of data the tool requires which allowed for scoring in the EMR. This was chosen over shorter screening tools, to allow for sufficient data collection from the electronic medical record and because it is already used in the setting the study was conducted in. The bed-side score was considered as the ‘gold standard’ for diagnosing ‘risk’ in this study, when compared with the EMR tool score. There are many other tools used for screening malnutrition risk that are also considered as highly valid, including the Malnutrition Screening Tool (MST) and Subjective Global Assessment (SGA) screening tool [3,9]. The PG-SGA guidelines suggest the level of need for further assessment according to scoring ranges: a score of 0–1 indicates that there’s ‘no nutrition intervention required at this time’, a score of 2–3 indicates that the ‘patient education potentially needed but no nutrition intervention’, while a score of 4–8, ‘requires intervention by dietitian to assess malnutrition in conjunction with nurse/physician as indicated by scored symptoms’, a score of >9 indicates a critical need [23]. Documentation in the EMR is both a combination of information verified with the patient, and information entered as a result of observation by health professionals, whereas the bed-side score is based only on the discussion with the patient.

The researchers scoring the EMR only considered that an item was “present” if EMR text indicated the patient had relevant symptoms with 100% certainty. No assumptions were made for symptoms that were not clearly described in the clinical notes (i.e., no documentation on loss of appetite meant this symptom was assumed not to be present, therefore no point was assigned under this criteria). A 10% sample of the EMR scores were cross-checked by a third researcher (LA) with 100% agreement. Demographic information as collected at admission was recorded, including sex, age (in years) and whether the patient lived in the larger rural township or smaller surrounding rural communities. Length of stay (LoS) in days was also collected.

### 2.2. Analysis

The sample size was determined pragmatically within the resource limitations of this exploratory pilot study. Statistical analysis was performed with Stata SE Version 15 (StataCorp LLC, College Station, TX, USA) [25]. The accuracy of the EMR score compared to the bedside score was assessed through area under the receiver operating characteristic (ROC, AUC), and sensitivity and specificity defined based on the cut-off recommended for the PG-SGA (score > 4) [24]. Estimates are presented along their 95% confidence intervals (CI). The optimal cut-off score based on the ROC was selected as the one with the largest Youden index (YI); i.e., the cut-off that maximises (sensitivity + specificity − 1) [26].

## 3. Results

Fifty in-patients that met the eligibility criteria were recruited in a rural hospital setting over a 12-week period (25% of eligible admissions in this service during the study timeframe). Table 1 shows the demographic and clinical characteristics of the sample. The mean age of the participants was 70.6 years (Standard Deviation (SD):14.9) and 62% were female. According to the EMR documentation, 32% of participants had one or more of the health conditions considered to increase malnutrition risk by the PG-SGA, along with 28% reporting loss of appetite during their admission. The mean bedside score was 8.0 (range 1–29) and the mean EMR score was 7.9 (0–22). Nearly three-in-four participants (74%) were rated at risk of malnutrition and needing further assessment by a dietitian (score ≥ 4) by the bedside screen and 80% were rated as at risk by the EMR score.

The EMR score showed a moderate accuracy compared to the usual bedside scoring method; indicated by an Area Under the Curve (AUC) of 0.74 (95% CI: 0.59–0.89; Figure 1).

Table 2 shows the sensitivity and specificity of the EMR score compared to the bed-side screen at different cut points. The sensitivity of the EMR score based on the cut-off proposed by the PG-SGA for the bedside score (>4) was 86.5% while the specificity was 38.5%. The Youden Index was maximized at cut point 7 with a sensitivity of 67.6 and a specificity of 76.9% and 70% of the patients correctly classified.

## 4. Discussion

This exploratory study aimed to understand whether the PG-SGA malnutrition screening tool when applied to EMRs, can accurately identify patients at risk of malnutrition, compared to screening methods at the patient bedside. This tool was chosen by the research team due to the the high scoring range which enabled enough data to be collected from the EMR to assess accuracy. As this study is one of the first to explore the accuracy of a bed-side tool when applied to the EMR, we acknowledge that research into other tools is also needed. This analysis showed that screening of clinical notes in the EMR had a good sensitivity and specificity for patients who are at risk of malnutrition. This suggests that EMR screening may be a promising approach for resource limited settings, where uptake of mandatory screening is low and where it is essential at risk patients are identified, prioritised, and assessed by a clinical dietitian. Our sample, commensurate with the rural hospital where the study was conducted, was 25% of all admissions over the timeframe, generating a representative sample in this context. For example, in data from all annual admissions, from the same setting, we identified a similar rate of patients at risk of malnutrition being 80% in this sample, compared to 77% in the larger sample from more than 500 admissions [17]. The large sample was also 60.2% female, with a mean age of 70.6 years, which is similar to the population presented in this study [17]. Our sample was also 70.6 years on average, with a LoS of 5 days. Our findings showed that the EMR screen was more likely to pick up patients at risk of malnutrition, which is ideal in the clinical setting. For example, for patients scoring 7 or higher, the EMR screen would detect 70% of patients at high risk with relatively high sensitivity and specificity. Although imperfect, studies in larger samples may provide more clarity on the accuracy of the tool and how patients scoring above 7 may be better identified. This is the first study to investigate the accuracy of traditional bed-side malnutrition screening when applied to EMRs, globally. There are large gaps in this area of research, likely due to the rapid and increasing change in the adoption of technology, especially in healthcare settings in developed countries such as Australia.

Health systems are increasingly transitioning to electronic record keeping due to the opportunities for technology to improve the care and coordination of patients, especially those with chronic and complex conditions [27]. This is despite the recognition that medical staff are frustrated by the level of administrative burden that electronic systems create, and the perspective that more time is spent entering data, than on patient interactions [28]. An explorative study published in 2021 showed that including a screening tool in-patient’s EMRs, with a prompt for nurses to complete it on admission, led to improved compliance to screening policy and referrals to dietitians [29]. Despite there being only minimal exploration of the use of EMRs in dietetic care, the more onerous method of paper-based medical record screening has been used previously in research to understand malnutrition risk [12].

The use of EMRs has previously been shown to improve dietetic care [30]. For example, an assessment of the impact on EMRs on dietetic service provision found that the proportion of nutrition diagnoses addressed and resolved during admission increased from 20.0% to 34.0% following EMR implementation, which is likely due to clearer documentation in comparison to previously handwritten notes [30]. Our study also offers further scope for exploration on how dietetics services can be improved through exploring potential uses of EMRs. As an example, automated text screening of EMRs, underpinned by validated tools (as shown here), could also enhance this process by flagging patients at risk of malnutrition, not only to hospital staff but their GP or private dietitian to enhance the continuum of care. Importantly, our study shows that there may also be potential problems with the use of EMRs to automatically screen in-patients. For example, there were a few ‘false’ positives in our sample, where patients were identified by the EMR score to be at risk, but the bed-side screen indicated that they were not at increased risk. This could potentially lead to inappropriate use of resources, and dietitians would still need to use clinical judgement when first approaching the patients to determine if a full nutrition assessment and treatment plan is needed. This could also be due to the difference between patient perceptions (that inform the bed-side score) versus the clinical documentation (which is based on both the patient’s report and health professional observations). It may be that patients answered questions at the bedside in a manner that meant they scored lower, whereas the clinical notes, documented by multiple different health professionals, may have provided more information leading to the EMR score. Other tools, that enable enough data collection from the EMR to generate a score, should also be explored to understand the applicability of other tools in this context.

The translation of clinical data to clinician knowledge to improve care is one of the current gaps of the transition to EMRs, and across the settings where an individual may receive care over any given time-frame [14]. For example, our study has shown that screening of EMRs has moderate sensitivity for picking up patients with higher scores, which would be useful in the clinical setting. Further research could seek to understand how simple programs that use auto-mated text recognition, could be used to automatically screen notes, scoring based on the PG-SGA and generate automated referrals to dietitians. This would reduce administrative burden on staff who would normally undertake the screening and ensure each patient receives a mandatory screen.

## 5. Strengths and Limitations

A strength of this study is that one trained researcher undertook the bed-side screening therefore ensuring diagnostic consistency and avoiding suspicion bias possibly caused by different interviewers, noted in other similar studies comparing specificity and sensitivity and clinical tools [26]. A limitation of this study is that it was undertaken in a small sample, although representative of the eligible population within the rural context. In further studies, different indexes of agreement could also be applied. A further limitation is that the ‘reason for admission’ and diagnosis were not entered in a standardised format into the EMRs by clinical staff in this setting, meaning this data was of poor quality and unable to be collected. We also did not compare documentation at the symptom level, between the EMR and the bedside, due to small numbers (e.g., only 1 patients had mouth sores documented in their file) A further limitation is that we did not measure and compare time taken to apply the bedside screen, versus the EMR screen, and this should be a focus in further research, although it can be assumed that collecting information from typed clinical notes that are entered into templates is quicker to collect than hand-written paper-based entries. It is highly likely that this process may have saved time as the researchers were able to screen EMRs at the desk, instead of travelling to the ward and discussing the screening questions with the patients, but the magnitude of this was not explored here. Further, patient weight and potentially other symptoms, were also frequently not documented in this sample, which meant that scoring of the EMR may have been under-estimated.

## 6. Conclusions

This exploratory study shows that applying the PG-SGA screening tool to the text in EMRs has moderate sensitivity and specificity for identifying patients at a risk of malnutrition. This provides evidence that the EMR screen has moderate accuracy in identifying patients at risk of malnutrition and that there is justification for further exploration into EMR screening to assist with identifying and triaging patients for dietetics referrals especially in rural settings where resources may be limited. This also warrants further exploration into how applying traditional bed-side screening tools perhaps through technology, to an EMR, could enable faster identification of high-risk patients. This could be an important strategy to address the low uptake of mandatory screening in hospitals, in Australia and around the world.

## Figures and Tables

**Figure 1 nutrients-14-00205-f001:**
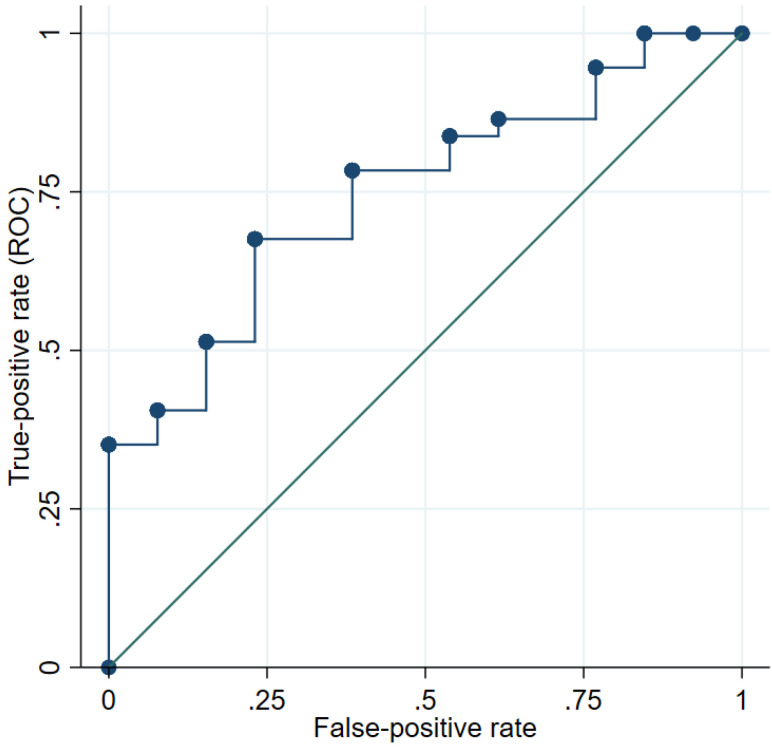
Receiver Operating Characteristic (ROC) curve for the malnutrition PG-SGA score based on the EMR review compared to PG-SGA bed-side screen.

**Table 1 nutrients-14-00205-t001:** Demographics and clinical characteristics of the patients.

Patient Characteristic	Sample *n* = 50
Mean Age (SD) (years)	70.6 (14.9)
Median LoS (range) (days)	5.0 (3–19)
Female *N* (%)	31 (62%)
Proportion residing in the main rural township *N* (%)	39 (78%)
**Nutrition symptoms documented in record *N* (%)**	
Presence of malnutrition risk conditions (cancer, AIDs, pulmonary cachexia, cardiac cachexia, open wound or trauma)	16 (32%)
Loss of appetite	14 (28%)
Early satiety	1 (2%)
Poor taste	1 (2%)
Mouth sores	1 (2%)
Nausea	8 (16%)
Constipation	16 (32%)
Vomiting	6 (12%)
Swallowing difficulties	1 (2%)
**Malnutrition screening results**	
Mean bedside score (range)	8.0 (1–29)
Mean EMR score (range)	7.9 (0–22)
At risk of malnutrition based on bedside score (95% CI) (PG SGA score >4)	74% (59–84%)
At risk based on EMR score (95% CI) (PG SGA score >4)	80% (66–89%)

‘SD’ standard deviation, ‘EMR’ electronic medical record, ‘PG SGA’ Patient Guided Subjective Global Assessment, ‘CI’ confidence interval.

**Table 2 nutrients-14-00205-t002:** Sensitivity, specificity, Youden Index at different cut off points of the EMR-screen PG-SGA.

Cutpoint (Ideal > 4)	Sensitivity	Specificity	Correctly Classified	Youden Index
(≥0)	100.0%	0.0%	74.0%	0.0
(≥1)	100.0%	7.7%	76.0%	0.08
(≥2)	100.0%	15.4%	78.0%	0.15
(≥3)	94.6%	23.1%	76.0%	0.18
(≥4)	86.5%	38.5%	74.0%	0.25
(≥5)	83.8%	46.2%	74.0%	0.30
(≥6)	78.4%	61.5%	74.0%	0.40
(≥7)	67.6%	76.9%	70.0%	0.45
(≥8)	51.4%	76.9%	58.0%	0.28
(≥9)	51.4%	84.6%	60.0%	0.36
(≥10)	40.5%	92.3%	54.0%	0.33
(≥11)	35.1%	100.0%	52.0%	0.35
(≥12)	27.0%	100.0%	46.0%	0.27
(≥14)	18.9%	100.0%	40.0%	0.19
(≥15)	13.5%	100.0%	36.0%	0.14
(≥16)	10.8%	100.0%	34.0%	0.11
(≥18)	5.4%	100.0%	30.0%	0.05
(≥22)	2.7%	100.0%	28.0%	0.03
(>22)	0.0%	100.0%	26.0%	0.0

## Data Availability

We do not have ethics approval to share the data from this study.

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
