# Peer review of "Testing the Accuracy of a Bedside Screening Tool Framework to Clinical Records for Identification of Patients at Risk of Malnutrition in a Rural Setting: An Exploratory Study"

_nutrients, 2022, doi:10.3390/nu14010205_

Round 1

Reviewer 1 Report

This exploratory study investigated the accuracy of the PG-SGA based on the EMR for the PG-SGA assessed at the bedside in patients admitted to rural health services. This investigation addressed clinically relevant issues as nutritional screening is an essential step for all healthcare settings, despite the limited human resources, especially in rural settings. I have a few comments that need to be addressed.

#1 Introduction: In my understanding, the PG-SGA is a nutritional assessment tool rather than a screening tool (https://pubmed.ncbi.nlm.nih.gov/12122555/). The use of this tool for nutritional screening can be acceptable, but it may be more complex and time-consuming than other traditional screening tools such as MNA-SF or MUST. The authors should state why the PG-SGA as a screening tool had priority to study over other more easy-to-utilise and straightforward tools.

#2 L115-116: It needs to be stated whether the independent researchers who assessed the patients’ EMR were blinded to the bedside PG-SGA score.

#3 L157: The reason for admission for the study participants should be described to verify the generalizability of the study results.

#4 L178-181: It seems strange: according to Table2, the maximum Youden Index was found in cutpoint 7 (0.45) rather than 4 (0.25). The authors should clarify this discrepancy.

#5 Discussion: As suggested in comment#1, the priority of the PG-SGA over other tools should be discussed.

#6 L284: One major limitation of this study is that some information might not be recorded in EMR; thus, the results could be underestimated. This point was partially stated in L296-298, but it may not be limited to bodyweight issues. The authors would clarify how this problem affected the study results.

Author Response

Thank you reviewer 1 for your very helpful review.

#1 Introduction: In my understanding, the PG-SGA is a nutritional assessment tool rather than a screening tool (https://pubmed.ncbi.nlm.nih.gov/12122555/). The use of this tool for nutritional screening can be acceptable, but it may be more complex and time-consuming than other traditional screening tools such as MNA-SF or MUST. The authors should state why the PG-SGA as a screening tool had priority to study over other more easy-to-utilise and straightforward tools.

Response: Thank you for raising this important point. We have now included why we have used this tool over the less comprehensive tools in the methods. “The researchers selected the PG-SGA tool to explore in this study due to the large amount of data the tool requires which allowed for scoring in the EMR. This was chosen over shorter screening tools, to allow for sufficient data collection from the electronic medical record.’        Reference: lines 131-133 of the methods section.

#2 L115-116: It needs to be stated whether the independent researchers who assessed the patients’ EMR were blinded to the bedside PG-SGA score.

Response: We have now ensured that this is stated in the methods              Reference: line 116 of the methods section.

#3 L157: The reason for admission for the study participants should be described to verify the generalizability of the study results.        

Response: We agree with the reviewer that this information is helpful. However, the reasons for admission are entered by nursing staff and were not standardised reasons. This data was therefore considered of poor quality and was not included here. We have now included this in our limitations section and agree with the reviewer. ‘A further limitation is that the ‘reason for admission’ and diagnosis were not entered in a standardised format into the EMRs by clinical staff in this setting, meaning this data was of poor quality and unable to be collected’.         Reference: line 292-295.

#4 L178-181: It seems strange: according to Table2, the maximum Youden Index was found in cutpoint 7 (0.45) rather than 4 (0.25). The authors should clarify this discrepancy. 

Response: The cut-off of four is relevant as this is the cut-off for the PG-SGA and a score of 4 or above indicates need for dietetics intervention. We have clarified this in the results.    Reference: lines 183-184.

#5 Discussion: As suggested in comment#1, the priority of the PG-SGA over other tools should be discussed.

Response: We agree with the reviewer. The PG-SGA was chosen in this context because of its large scoring range and was chosen opportunistically by the research team. Other tools should also be explored in a similar way, and we have added this perspective to the discussion.   Reference: lines 191-195, 267-270

#6 L284: One major limitation of this study is that some information might not be recorded in EMR; thus, the results could be underestimated. This point was partially stated in L296-298, but it may not be limited to bodyweight issues. The authors would clarify how this problem affected the study results.           

Response: We agree with the reviewer and have clarified this in our limitations.              Reference: lines 309-311

Reviewer 2 Report

This study investigate the accuracy of a malnutrition risk score obtained by applying the PG-SGA screening instrument to the clinical text in electronic medical records, relative to the PG-SGA instrument applied at the bedside, in identifying rural in-patients at an increased risk. of malnutrition. Lines 118 to 136 I think are redundant since PG-SGA is a well known test that does not need much explanation. In Table 1, I would add the diagnoses causing admission and medical history. The data in Table 1 referring to Nutrition symptoms documented in record N (%) would be put in another table with two columns: one from PG-SGA bedside and another with PG-SGA from electronic medical records. I think that the determination of the cut-off point is not the most interesting test. I think that the Kappa index of agreement or the Bland and Altman test should be calculated. On the other hand, this study seems to me not very reproducible because it depends in each hospital on the data that are usually collected in the electronic medical records. This may differ from hospital to hospital or even within the same hospital from doctor to doctor. 

Author Response

Thank you to reviewer 2 for taking the time to review our manuscript. We really appreciate your helpful comments.

#1 This study investigate the accuracy of a malnutrition risk score obtained by applying the PG-SGA screening instrument to the clinical text in electronic medical records, relative to the PG-SGA instrument applied at the bedside, in identifying rural in-patients at an increased risk. of malnutrition. Lines 118 to 136 I think are redundant since PG-SGA is a well-known test that does not need much explanation.          

Response: Although we’d like to present as much information as possible on the tool, we do agree with the reviewer and have reduced this size of this section.              Reference: lines 120-123

#2 In Table 1, I would add the diagnoses causing admission and medical history. The data in Table 1 referring to Nutrition symptoms documented in record N (%) would be put in another table with two columns: one from PG-SGA bedside and another with PG-SGA from electronic medical records.          

Response: We agree with the reviewer that this information is helpful. However, the reasons for admission are entered by nursing staff and were not standardised reasons. This data was therefore considered of poor quality and was not included here. We have now included this in our limitations section and agree with the reviewer.

We have also only included the data from the EMR in table 1 as we have not made comparisons between the individual items, as this would not offer additional knowledge in a sample of this size. We will include this in future studies and have put this in our limitations.

The limitations now state: A further limitation is that the ‘reason for admission’ and diagnosis were not entered in a standardised format into the EMRs by clinical staff in this setting, meaning this data was of poor quality and unable to be collected. We also did not compare documentation at the symptom level, between the EMR and the bedside, due to small numbers (e.g. only 1 patients had mouth sores documented in their file).           Reference: lines 292-304 and table 1.

#3 I think that the determination of the cut-off point is not the most interesting test. I think that the Kappa index of agreement or the Bland and Altman test should be calculated. On the other hand, this study seems to me not very reproducible because it depends in each hospital on the data that are usually collected in the electronic medical records. This may differ from hospital to hospital or even within the same hospital from doctor to doctor.

Response: Thank you for this suggestion, due to the small sample and aims of this exploratory study and based on other similar literature, the Youden index was selected in this study. We agree that this information would be useful and have added it to our suggestions for further research.               Reference: line 299 of the limitations.

Round 2

Reviewer 1 Report

I have no further comment.

Author Response

Thank you kindly for reviewing our paper. 

kind regards,

Dr Alston